# Impact of prior SARS-CoV-2 infection on perioperative cardiac, pulmonary and neurocognitive complications in older patients: Study protocol for an observative case control study

**Dayana Abramova, Paula Marie Haase** , **Anne-Marie Just, Sandra Frank** , **Thomas Saller**  *

Department of Anaesthesiology, LMU University Hospital, LMU Munich, Campus Großhadern, Munich, Germany

* tsaller@med.lmu.de

## Abstract

### Background

Postoperative delirium is considered a serious complication in older patients. Older patients often suffer from several concomitant diseases. The reduced physical condition can increase the risk of cardiac, pulmonary and neurocognitive complications during and after surgery. SARS-CoV-2 infection primarily affects the respiratory tract but can also damage other organ systems such as the heart and brain. Given the wide range of pulmonary, cardiac and neurocognitive complications caused by SARS-CoV-2, these risks must be given special consideration during planned surgical procedures. Both surgical procedures and anesthesia are risk factors for postoperative complications in themselves. The specific impact of prior SARS-CoV-2 infection on perioperative complications in older patients has not been sufficiently researched. The aim of this study is to understand how a previous SARS-CoV-2 infection influences the occurrence of perioperative complications.

### Methods

In this case-control study, the data of patients over 60 years of age undergoing elective surgery are analyzed. Subjects are divided into two groups based on their SARS-CoV-2 infection status: those with a documented previous infection and those without. Confirmation of infection will be based on written evidence and anamnestic information. The primary endpoint of the examination is the occurrence of delirium within the first five postoperative days. In addition, further cardiac, pulmonary and neurocognitive complications are recorded in the perioperative period. The occurrence of postoperative delirium is recorded during the daily ward round in the first

**Data availability statement:** All data are in the manuscript and/or supporting information files.

**Funding:** The author(s) received no specific funding for this work.

**Competing interests:** The authors have declared that no competing interests exist.

**Abbreviations:** 3DCAM, 3-dimensional confusion assessment method; 4AT, alertness, attention, acute change and abbreviated mental test-4; ACE2, angiotensin-converting-enzyme-2-receptors; ALI, acute lung injury; ARDS, acute respiratory distress syndrome; ASA, American Society of Anesthesiologists' Physical Status; BDSG, Bundesdatenschutzgesetz (Federal Data Protection Act); BMI, body-mass index; CARDS, COVID-19-associated Acute Respiratory Distress Syndrome; COVID-19, coronavirus disease 2019; CRP, C-reactive protein; eCRF, electronic Case Report Form; $FiO_2$, fraction of inspired oxygen; GCP, good clinical practice; GI, gastrointestinal; ICU, intensive care unit; KAS, Krankenhaus-Informations- und Verwaltungssystem (hospital information and administration system); LMU, Ludwig-Maximilians-Universität; MAP, mean Arterial Pressure; MOCA, Montreal Cognitive Assessment; NarkoPrämed, Narkose Prämedikation (anesthesia premedication); NRS, numerical Rating Scale; $PaO_2$, partial pressure of arterial oxygen; $PaCO_2$, partial pressure of arterial carbon dioxide; PCPF, Post-COVID pulmonary fibrosis; PCR, polymerase chain reaction test; PEEP, positive end-expiratory pressure; Pmean, mean pressure; PONV, postoperative nausea and vomiting; Ppeak, peak pressure; REDCap, research electronic data capture; RKI, Robert Koch Institut; SARS-CoV-2, severe acute respiratory syndrome coronavirus 2; $SpO_2$, oxygen saturatios; SSRI, selective serotonin reuptake inhibitor; TIVA, total intravenous anesthesia; VT, tidal volume

five days after surgery. The 3DCAM test and the 4AT are used for this purpose. In addition, the CAM-ICU will be used in the intensive care unit. The recruitment will include 266 patients. Statistical analyses will be performed to determine the correlation between a previous SARS-CoV-2 infection and the observed clinical outcomes.

## Discussion

The results of this study will provide new insights into the impact of prior SARS-CoV-2 infection on perioperative complications in older patients undergoing elective surgery.

## Trial registration

Deutsches Register Klinischer Studien: DRKS00034861.

## Background

Investigating the effects of surgical interventions under general anesthesia on the human organism is a crucial focus of scientific research. In view of the globally aging population, there is great interest in ensuring safe and high-quality surgical care for the older patients.

Older people are often affected by multiple comorbidities. The patient's reduced physical condition can increase the risk of cardiac, pulmonary and neurocognitive complications during and after surgery. In addition, age-related changes in the brain lead to reduced reserve capacity [1]. Among older individuals, this can increase the risk of postoperative delirium. Postoperative delirium is considered a serious complication in older patients and is also the most common after surgery and associated with prolonged need for care. The strongest predictor of postoperative delirium is pre-existing cognitive dysfunction [2]. However, the frequency of postoperative delirium in older patients with cognitive deficits can be reduced by targeted measures [3]. Consequently, it is important to identify patients with risk factors and adapt perioperative management accordingly.

The SARS-CoV-2 pandemic has brought additional challenges in hospital and perioperative management [4]. COVID-19 is generally known to affect the respiratory tract. A SARS-CoV-2-infection can also have effects on other organ systems such as the heart, brain, liver and kidneys [5].

Within months of a COVID-19 disease, functional limitations of the lungs could be detected. This was demonstrated by examinations of lung function, physical fitness, oxygen supply during physical exertion and radiological findings. For this reason, systematic follow-up care is recommended, especially after severe courses of COVID-19 [6].

The effects of SARS-CoV-2 on the lungs can be devastating, depending on the course of the disease, age and comorbidities. The impairment of lung function also increases the likelihood of cardiac and vascular complications. An international initiative proposed that surgery should be delayed for at least 7 weeks following a

SARS-CoV-2 infection [7]. However, there remains limited knowledge regarding the long-term effects of Covid-19 disease on perioperative complications and morbidity.

As the virus can also damage the brain, neurological complications have gained significant attention. The endothelial cells of the brain express ACE2 receptors and are therefore susceptible to infection with SARS-CoV-2 [5,8]. Known long-term neurological symptoms include smell and taste disorders, headaches, muscle pain, memory problems, anxiety and sleep disorders. SARS-CoV-2 can also cause states of confusion (delirium) and agitation [9]. Older patients with pre-existing cognitive impairment are particularly at risk.

Given the multiple pulmonary, cardiac and neurocognitive complications caused by SARS-CoV-2, these must also be considered in upcoming surgical procedures. This is because both the surgical procedure and the anesthesia itself already represent risk factors for possible postoperative complications.

Currently, there is great uncertainty about the long-term consequences of SARS-CoV-2 infection in the perioperative setting. There is limited valid and generalizable data on the occurrence of perioperative complications in patients with previous SARS-CoV-2 infection, particularly in the older population. New studies are therefore needed to determine the long-term effects of SARS-CoV-2 infection on perioperative cardiac, pulmonary and neurocognitive complications in older patients.

## Methods and design

### Ethics statement

This study has been approved by the Ethics Committee of the Ludwig-Maximilians-Universität München (LMU Munich). The ethics committee reviewed and approved the study protocol under the reference number 23-0552. Informed written consent was obtained from all participants involved in the study. The study has been conducted in accordance with the Declaration of Helsinki and all applicable ethical regulations.

### Outcomes

The aim of the study is to investigate effects of a previous SARS-CoV-2 infection on the perioperative care and outcome of older patients with a focus on the incidence of postoperative delirium.

**Primary outcome measures.** The primary aim is to determine the incidence of postoperative delirium on one of the first five days after surgery in the case group compared to the control group. The endpoint will be considered met if postoperative delirium is detected with at least one of the applied assessment methods used.

**Secondary outcome measures.** Intraoperative ventilation settings will be evaluated. Additionally, the incidence of unplanned postoperative ventilation, oxygen therapy, ARDS, respiratory insufficiency, pneumonia, pulmonary embolism, pneumothorax, myocardial infarction, myocardial damage, cardiac death, cardiac arrhythmias, atrial fibrillation, stroke, acute renal failure and other serious postoperative complications are going to be analyzed.

### Number of participants

A priori test strength analysis was performed to estimate the required number of participants. This results in a total study population of 242 participants with a required sample size of 121 participants per group. To ensure the calculated minimum level of power with the possibility of dropouts this number is increased by 10%. This results in a final planned sample size of 266 participants. Recruitment started on 06.09.2023 with the approval of the ethics vote and is expected to end on 31.12.2024.

### Inclusion criteria

- patients aged 60 years or older

- patients with an elective surgical procedure lasting 1 hour or more

- patients undergoing surgery under general anesthesia

- patients capable of giving consent

- signed declaration of consent from the patient

**Exclusion criteria**

- emergency surgery

- patients with language barriers

- patients with preoperative delirium during hospital admission

- unwillingness to follow the study protocol

**Recruitment**

The selection of participants for the study will take place at the anesthesia outpatient clinic of the University Hospital Munich, LMU Munich. A structured approach will be followed to identify and include all patients who fulfill the defined inclusion criteria and do not meet any of the exclusion criteria.

**Consent**

Information regarding the study is provided both verbally and in writing during the preoperative anesthesia informed consent discussion. In this context, potential study participants are presented with an information sheet that explains the objectives, procedure, potential risks and benefits of the study in detail. Patients receive a copy of the information sheet for their personal use.

A detailed Covid history is also recorded. This includes the number and time of Covid infections, as well as the time between the last infection and the operation. In addition, postoperative pain, including pain therapy and postoperative nausea and vomiting are determined.

**Duration of the study**

The total duration of the study is planned to be 15 months.

**Study design**

This study follows a case-control design and is conducted both retrospectively and prospectively as a non-interventional observational study.

The first phase contains a retrospective analysis of an ongoing, non-interventional delirium study. The generated data will be included in the study. Additional participants are included to optimize the number of cases and to increase the statistical significance.

Patients who meet the specified inclusion criteria and do not meet any exclusion criteria are informed about the study during their anesthesiology consultation. After being provided with information and an appropriate reflection period, the participants give their written consent to participate by signing the consent formular.

All patients undergo routine frailty screening. This is performed by specially trained staff, who also perform the Mini-Cog test. In addition, all study participants undergo a Montreal Cognitive Assessment (MOCA) and the 3-dimensional Cognitive Assessment Method (3DCAM), which are a prerequisite for inclusion in the study. The MOCA test is used to further assess the patient's cognitive state, which can affect the postoperative outcome and the incidence of postoperative delirium.

Demographic data, the Clinical Frailty Scale, the frailty criteria according to Fried, the physical status according to the American Society of Anesthesiologists (ASA), the functional status, main diagnoses, relevant previous illnesses and the patient's current medication are recorded in an anamnesis form. The current medication is analyzed with a particular focus on specific drugs that may impact the study outcome. These include anticholinergics, antihypertensives, benzodiazepines, tricyclic antidepressants, selective serotonin reuptake inhibitors (SSRIs), hypnotics and opioids. The total number of drugs administered is also recorded in this context. This information is extracted from the NarkoPrämed system (IMESEO GmbH, Gießen, Germany).

In addition, a detailed history of smoking, chronic pain and COVID-specific medical history is obtained. Patients are interviewed about details of a SARS-CoV-2 infections and the dates of these infections. If the exact date of the last positive PCR test is unknown, participants are categorized into specific time intervals: less than 7 weeks, less than 3 months, less than 6 months, less than 1 year or more than 1 year. The COVID-related medical history also includes questions about the symptoms and their severity that occurred in the course of the COVID-19 disease. All presurgical data are summarized in Table 1.

The four primary symptoms of COVID-19 disease are specifically recorded. These are cough, fever, runny nose and disturbance of the sense of smell and/or taste [10]. In addition, information about possible complications of the COVID disease, currently still existing symptoms, the possible presence of a long-COVID or post-COVID syndrome, vaccination status and past hospitalization are collected. Supporting S1 Fig presents the SPIRIT schedule of the study.

The study participants are then divided into two cohorts. The case group includes of patients over 60 years of age undergoing elective surgery under general anesthesia lasting more than one hour and a previous SARS-CoV-2 infection. The control group includes of patients over 60 years of age undergoing elective surgery under general anesthesia lasting more than one hour without a previously diagnosed SARS-CoV-2 infection. Both groups have an equal number of patients.

Routine laboratory parameters will be recorded as part of the preoperative surgical admission and examination. All laboratory parameters relevant to the study will be extracted and documented from the patient's electronic file, if available. These include C-reactive protein (CRP), hemoglobin, creatinine, interleukin 6, urea, glucose, sodium and potassium.

Data on the surgical procedure is also collected, such as the urgency of the operation, the surgical specialty, the surgical technique, the duration of the operation, anesthesia technique, airway management, whether epidural anesthesia was

**Table 1. Presurgical data.**

| Presurgical data | Items |
|---|---|
| Demographic data | age, gender, height, weight, BMI |
| Risk classification | Clinical Frailty Scale, frailty criteria according to Fried, ASA, functional status, PONV risk |
| Risk factors | smoking history, alcohol consumption, chronic pain |
| Diseases | diagnosis, previous illnesses |
| Long-term medication | anticholinergics, benzodiazepines, tricyclic antidepressants, SSRIs, hypnotics, opioids, other, number of medications |
| COVID-specific medical history | previous SARS-CoV-2 infection, time of last SARS-CoV-2 infection, symptoms, severity of symptoms, complications, symptoms still present, long-COVID or post-COVID, SARS-CoV-2 vaccination status, time of last vaccination |
| Routine laboratory parameters | C-reactive protein, haemoglobin, creatinine, interleukin 6, urea, glucose, sodium, potassium |
| Surgical discipline | Orthopedics, Trauma surgery, Gynecology & Obstetrics, Upper GI tract, Lower GI tract, Hepato-biliary, Endocrine surgery and pancreas, Head and neck, Breast surgery, Cardiac, Thoracic (esophagus), Thoracic (lung and other), Vascular, Urology and kidney, Plastic/dermatologic surgery, Other |
| Cognitive impairment | MOCA-Test, Mini Cog-Test |

performed, the duration of anesthesia and the extent of blood transfusions. Additionally, preoperative parameters from the anesthesia protocol will be collected, such as oxygen partial pressure, carbon dioxide partial pressure, inspiratory oxygen concentration (if oxygen was administered), pH, respiratory rate, blood pressure, heart rate, temperature and oxygen saturation. These parameters are recorded both during and after surgery. Intraoperative ventilation parameters to be included in the data collection are ventilation mode, driving pressure, plateau pressure, positive end-expiratory pulmonary pressure, compliance, tidal volume, and mean arterial pressure (MAP). All intraoperative complications including arrhythmias, hypotension, drop in oxygen saturation, blood loss and other events are documented. Anesthesia and surgery treatment data will be extracted from the hospital's KAS data information system using surgical and anesthesia protocols. All data is recorded before, during and after anesthesia.

After surgery, any relevant clinical data and complications that occur during the first five postoperative days are documented. Primarily, delirium and potential postoperative complications, such as nausea and vomiting (PONV) and pain are recorded. The Apfel-Score is used to estimate the postoperative PONV risk. The following factors each represent one point: female gender, non-smoker, PONV history and postoperative opioid administration. The higher the score, the higher the risk of PONV [11]. Pain intensity is assessed using the Numerical Rating Scale (NRS, 0 = no pain, 10 = worst pain imaginable), and the administered pain therapy is also documented [12]. The PONV Intensity Scale according to Wengritzky quantify postoperative nausea and vomiting [13].

Further documentation includes intensive care unit (ICU) admission, patient orientation (time, place, person, situation), prolonged or newly initiated oxygen therapy and the need for mechanical ventilation postoperatively. Postsurgical complications occurring on the day of surgery are reviewed in patient files, particularly the anesthesia protocol, using the Nu-DESC (Nursing Delirium Screening Scale) and anesthesia progress observations (AVB). All treatment data are summarized in Table 2.

The occurrence of postoperative delirium is assessed during daily rounds by trained staff in the first five days after the operation, in accordance with the European guideline. The 3DCAM test and 4AT are used for this purpose.

Additionally, the CAM-ICU (Confusion Assessment Method for the Intensive Care Unit) is routinely used by medical staff in the intensive care unit. All three instruments are validated and reliable screening methods for detecting delirium [14,15].

Withdrawal from the study happens may occur after the patient revokes consent, the planned operation is canceled, or if the study protocol is violated. All study participants are visited on all five postoperative days. To determine the incidence of delirium and other complications, such as cardiac and pulmonary, only patients who have spent at least one night on a surgical ward after surgery and who have undergone at least one postoperative screening will be included. Patients who do not complete these criteria are excluded from the statistical analysis. This methodological approach ensures a careful and representative analysis of postoperative events.

A recruitment period of approximately 16 months is planned to include 266 patients. A dropout rate of 10% of the recruited patients will be tolerated. If the dropout rate exceeds this threshold, further recruitment will be undertaken. The conduct of this study was guided by the recommendations of the STROBE checklist (Strengthening the Reporting of Observational Studies in Epidemiology) to ensure transparency and scientific rigor. All relevant points of the checklist were considered in the planning and conduct of the study to maximize the quality and reproducibility of the results.

## Documentation

The data is initially recorded and documented on paper and in an electronic database. This process is conducted by the doctors responsible for the study. The data is backed up and stored on a server located in Germany.

## Data monitoring

This study implements a monitoring concept to ensure data quality and safety in accordance with the guidelines for good clinical practice (GCP). Regular training of all personnel involved in the study, regular reviews for missing or inaccurate

data and regular team meetings are conducted. The training, monitoring and meetings are intended to ensure a high level of protocol compliance and data quality as well as patient safety and rights.

### Risk-benefit assessment

This study is designed as a non-interventional observational study. No adverse effects on the participants are anticipated. Potential adverse events will be carefully monitored both intraoperatively and postoperatively, providing the opportunity to initiate appropriate interventions in a timely manner. Postoperative safety is enhanced through additional visits by the study team, which consists of study physicians and doctoral students. Regarding to data protection, the collection and storage of data is pseudonymized.

### Termination criteria

Consent to participate in the study may be withdrawn at any time, without providing reasons and without negative impact on further medical treatment. In the event of withdrawal from the study, all personal data already collected will be deleted immediately and completely.

### Statistical analyses

Participants will be divided into two cohorts at the start of the study, depending on previous SARS-CoV-2 infection. The case group consists of patients over 60 years of age undergoing elective surgery under general anesthesia lasting more than one hour and a previous SARS-CoV-2 infection. The control group consists of patients with the same characteristics but without a history of SARS-CoV-2 infection. Both groups include the same number of patients.

**Table 2. Treatment data.**

| Treatment data | Items |
|---|---|
| Surgery | elective, emergency |
| Surgical technique | one-lung ventilation during the procedure, open abdominal surgery, open thoracic surgery, laparoscopic surgery, thoracoscopic surgery, laparoscopically or thoracoscopically assisted, robot-assisted surgery (e.g. daVinci surgery), other |
| Duration of surgery | start of surgery, end of surgery |
| Anesthesia | total intravenous anesthesia (TIVA), general (volatile), general (balanced) |
| Airway management | endotracheal tube, endobronchial blocker (tracheostomy), double-lumen endotracheal tube, laryngeal mask |
| Epidural anesthesia | none, thoracic, lumbar |
| Duration of anesthesia | start of anesthesia, end of anesthesia |
| Hydration balance | transfusion, blood loss |
| Preoperative parameters | $PaO_2$, $PaCO_2$, $FiO_2$, $etCO_2$, pH value, respiratory rate, blood pressure, heart rate, temperature, $SpO_2$ |
| Intraoperative parameters | $PaO_2$, $PaCO_2$, $FiO_2$, $etCO_2$, pH value, respiratory rate, blood pressure, heart rate, temperature, $SpO_2$, ventilation mode, Pmax, plateau, PEEP, compliance, respiratory minute volume, VT, MAP, complications |
| Postoperative parameters | $PaO_2$, $PaCO_2$, $FiO_2$, pH value, respiratory rate, blood pressure, heart rate, temperature, $SpO_2$, hemoglobin |
| Postoperative care | admission to the intensive care unit, reason for admission to the intensive care unit, occurrence of nausea and vomiting, pain intensity, need for oxygen therapy or mechanical ventilation, reason for oxygen therapy and mechanical ventilation, complications during the first 5 postoperative days |
| Postoperative pain therapy | opioids, peripheral analgesics, regional analgesics, other |
| Postoperative delirium screening | on postoperative days 1–5 using 3DCAM, 4AT and CAM-ICU in the intensive care unit |

The results are analyzed by comparing the case group with the control group using appropriate statistical methods. When analysing the data, we will use statistical methods to take confounders into account.

## Discussion

This study investigates the relationship between SARS-CoV-2 infection and the occurrence of postoperative delirium and other complications as for example reported for the cardiovascular system [16]. Despite careful planning and execution of the study, there are several potential sources of error that could affect the validity and reliability of the results.

In this study, a previous SARS-CoV-2 infection among subjects is confirmed through written evidence. In the cases where written evidence is unavailable, a SARS-CoV-2 infection may be confirmed based on the patient's history, provided the information is deemed adequate and credible. This method harbors several risks. Firstly, recall bias may occur, as patients might be unable to accurately recall whether and when they were infected with SARS-CoV-2. Secondly, asymptomatic infections could remain undetected, leading to underreporting of SARS-CoV-2 cases. These inaccuracies may distort the analysis of the relationship between prior COVID-19 and postoperative delirium and lead to erroneous conclusions.

Performing the same delirium screening on five consecutive postoperative days carries the risk of a learning effect among participants. Repeated testing might enable patients to familiarize themselves with the test questions, potentially resulting in improved performance over time, regardless of the actual presence or severity of delirium. This learning effect may reduce the sensitivity and specificity of delirium screening and lead to an under- or overestimation of delirium prevalence.

The heterogeneity of the patient population poses another challenge. Age differences and functional status may significantly influence the outcomes. Older patients are at higher risk of delirium due to age-related changes in the brain and a higher prevalence of comorbidities. In addition, differences in functional status may affect patients' ability to recover from surgery and could therefore skew delirium rates. An age- and function-specific analysis might be necessary to address this variability.

The MOCA test used for delirium detection in this study may be strongly influenced by the educational level of the participants. Individuals with higher education levels tend to score better on the MOCA, increasing the likelihood of underestimating cognitive impairment in less educated patients.

Ventilation data collected in the study may later help to identify correlations with perioperative complications.

In summary, it is essential to consider these potential sources of error when interpreting study results.

## Declarations

### Declaration of Helsinki

The study will be conducted in compliance with the protocol, GCP and the applicable regulatory requirements. The study will be performed in accordance with the recommendations guiding physicians in biomedical research involving human participants, adopted by the 18th World Medical Association General Assembly, Helsinki, Finland, June 1964, amended at the 48th World Medical Association General Assembly, Somerset West, Republic of South Africa, October 1996.

## Supporting information

**S1 Fig.  Study timeline with enrolment, intervention and assessment timepoints from baseline to postoperative day 5.**
(PDF)

## Author contributions

**Conceptualization:** Dayana Abramova, Sandra Frank, Thomas Saller.

**Data curation:** Dayana Abramova, Paula Marie Haase, Anne-Marie Just, Thomas Saller.

**Funding acquisition:** Thomas Saller.

**Investigation:** Dayana Abramova, Paula Marie Haase.

**Methodology:** Dayana Abramova, Thomas Saller.

**Project administration:** Anne-Marie Just, Thomas Saller.

**Resources:** Thomas Saller.

**Supervision:** Anne-Marie Just, Sandra Frank, Thomas Saller.

**Validation:** Anne-Marie Just.

**Writing – original draft:** Dayana Abramova, Paula Marie Haase, Anne-Marie Just, Thomas Saller.

**Writing – review & editing:** Dayana Abramova, Paula Marie Haase, Sandra Frank.

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
