## [Decision Letter · Decision Letter 0]

29 Oct 2024

PONE-D-24-39467Impact of prior SARS-CoV-2 infection on perioperative cardiac, pulmonary and neurocognitive complications in elderly patients: study protocol for an observative cohort studyPLOS ONE

Dear Dr. Saller,

Thank you for submitting your manuscript to PLOS ONE. After careful consideration, we feel that it has merit but does not fully meet PLOS ONE’s publication criteria as it currently stands. Therefore, we invite you to submit a revised version of the manuscript that addresses the points raised during the review process.

The manuscript is substantially well written, but some minor revisions are needed.

We look forward to receiving your revised manuscript.

Kind regards,

Prof. Raffaele Serra, M.D., Ph.D

Academic Editor

PLOS ONE

Journal Requirements:

4. Please include a caption for figure 1.

5. We note you have included a table to which you do not refer in the text of your manuscript. Please ensure that you refer to Tables 1 and 2 in your text; if accepted, production will need this reference to link the reader to the Table.

Reviewers' comments:

Reviewer's Responses to Questions

**Comments to the Author**

1. Does the manuscript provide a valid rationale for the proposed study, with clearly identified and justified research questions?

Reviewer #1: Yes

Reviewer #2: Yes

2. Is the protocol technically sound and planned in a manner that will lead to a meaningful outcome and allow testing the stated hypotheses?

Reviewer #1: Yes

Reviewer #2: Yes

3. Is the methodology feasible and described in sufficient detail to allow the work to be replicable?

Reviewer #1: Yes

Reviewer #2: Yes

4. Have the authors described where all data underlying the findings will be made available when the study is complete?

Reviewer #1: Yes

Reviewer #2: Yes

5. Is the manuscript presented in an intelligible fashion and written in standard English?

Reviewer #1: Yes

Reviewer #2: Yes

6. Review Comments to the Author

You may also provide optional suggestions and comments to authors that they might find helpful in planning their study.

Reviewer #1: The study titled "Impact of prior SARS-CoV-2 infection on perioperative cardiac, pulmonary, and neurocognitive complications in elderly patients" aims to investigate how previous SARS-CoV-2 infection affects the risk of perioperative complications in elderly patients undergoing elective surgeries. The study is important for Vulnerability of Elderly Patients, Long-term Impact of COVID-19, and Postoperative Delirium and Complications.

Overall, it addresses a critical gap in knowledge, focusing on how a prior SARS-CoV-2 infection affects surgical risks in a vulnerable population.

A minor question is about the duration. While the focus is on complications within the first five postoperative days, some complications might emerge later. Extending the follow-up period or adding a later check-in (e.g., at 30 days post-operation) could help identify delayed complications that are relevant to patient recovery.

Reviewer #2: The paper is very interesting in all parts and the methods are very interesting. There are some elements to improve: 1) the literature, in fact I suggest to include the following paper doi:10.2217/bmm-2020-0201.

2) it is necessary to improve the English.

7. PLOS authors have the option to publish the peer review history of their article (what does this mean? ). If published, this will include your full peer review and any attached files.

**Do you want your identity to be public for this peer review?** For information about this choice, including consent withdrawal, please see our Privacy Policy .

Reviewer #1: No

Reviewer #2: No

---

## [Author Response · Author response to Decision Letter 1]

16 Dec 2024

Reviewer #1

Thank you for taking the time to thoroughly review our manuscript and for providing such constructive feedback.

Comment #1.1

The study titled "Impact of prior SARS-CoV-2 infection on perioperative cardiac, pulmonary, and neurocognitive complications in elderly patients" aims to investigate how previous SARS-CoV-2 infection affects the risk of perioperative complications in elderly patients undergoing elective surgeries. The study is important for Vulnerability of Elderly Patients, Long-term Impact of COVID-19, and Postoperative Delirium and Complications. Overall, it addresses a critical gap in knowledge, focusing on how a prior SARS-CoV-2 infection affects surgical risks in a vulnerable population.

A minor question is about the duration. While the focus is on complications within the first five postoperative days, some complications might emerge later. Extending the follow-up period or adding a later check-in (e.g., at 30 days post-operation) could help identify delayed complications that are relevant to patient recovery.

Thank you very much for this comment. The decision to focus on the first five postoperative days, rather than extending the observation period to 30 days, is based on evidence indicating that the majority of postoperative complications, particularly delirium, occur within this timeframe. According to studies, the risk of postoperative delirium is highest within the initial three to five days following surgery, especially among patients in critical care settings.

This approach allows for the efficient allocation of resources. Extending the observation period to 30 days would increase the logistical and operational demands significantly, while contributing limited additional value, as complications occurring later are less frequent and often unrelated to the immediate perioperative period. By focusing on this critical window, the study aims to capture the most clinically relevant data and ensure high-quality monitoring of early postoperative outcomes.

Reviewer #2

Thank you very much for your encouraging comments and constructive feedback on our manuscript.

Comment #2.1

The paper is very interesting in all parts and the methods are very interesting. There are some elements to improve: 1) the literature, in fact I suggest to include the following paper doi:10.2217/bmm-2020-0201.

Thank you for suggesting the publication "Cardiovascular Disease as a Biomarker for an Increased Risk of COVID-19 Infection and Related Poor Prognosis" as a potential source for the study protocol. The article provides valuable insights into the pathophysiological mechanisms linking cardiovascular disease and COVID-19, including the roles of ACE2, systemic inflammation, and coagulopathies. The study protocol primarily aims to investigate perioperative risks, such as postoperative delirium and pulmonary outcomes, in elderly patients with a history of SARS-CoV-2 infection. The publication does not address surgical or perioperative contexts, nor does it explore complications like delirium, which is a central focus of the study. That said, the publication is an extremely valuable resource, and I believe it would be highly relevant to incorporate it into the Discussion section of the study’s final publication. In that section, we plan to analyze the cardiovascular comorbidities of our participants and discuss their impact on the severity of COVID-19, its systemic effects on the body, and consequently, the incidence of perioperative complications. The article’s detailed discussion of the interplay between cardiovascular disease and COVID-19 outcomes would provide a strong foundation for interpreting and contextualizing our findings.

Comment #2.2

2) it is necessary to improve the English.

We appreciate your rating and we have made numerous changes to improve the language.

JOURNAL Requirements

Adaption to the PLOS-ONE style requirements

The style requirements of PLOS ONE are now fulfilled and changes have been marked in yellow as specified in an extra manuscript, which has been named `Revised Manuscript`.

Data Availability Statement

The required Data Availability Statement has been added and can be found on page 17.

Ethic Statement

Our ethics vote has now been added to the methods as requested.

Caption for figure 1

A heading has been added for figure 1. Figure 1 has also been mentioned under `Supporting Information` as requested.

References to table 1 and 2

In the texts, the tables that follow the section have now been addressed.

Reference List

The references have not been changed, only the order has been adjusted according to the specified template.

---

## [Decision Letter · Decision Letter 1]

24 Feb 2025

PONE-D-24-39467R1Impact of prior SARS-CoV-2 infection on perioperative cardiac, pulmonary and neurocognitive complications in elderly patients: study protocol for an observative cohort studyPLOS ONE

Dear Dr. Saller,

Thank you for submitting your manuscript to PLOS ONE. After careful consideration, we feel that it has merit but does not fully meet PLOS ONE’s publication criteria as it currently stands. Therefore, we invite you to submit a revised version of the manuscript that addresses the points raised during the review process.

Thank you for taking the time to address the reviewer comments. **In addition to the reviewer comment below, please address the following:**

1) Methods - Please add reference to the STROBE checklist (https://www.strobe-statement.org/checklists/) and ensure the study is conducted according to this checklist to promote transparency and rigor.

2) Title - Please adjust ending to "study protocol for a case-control study" if this is accurate

3) Methods - Please discuss how COVID infection will be ascertained (i.e. asking patients versus laboratory evidence versus other)

4) Methods - Please discuss whether any propensity score matching or attempts of addressing confounders through adjustment will be conducted. These should ideally be included.

5) Please change the terminology of "elderly" to "older adults"

6) Please insert citations for the Numerical Rating Scale, PONV scales, and delirium (specify the tools in the actual Methods text)

We look forward to receiving your revised manuscript.

Kind regards,

Kiyan Heybati

Academic Editor

PLOS ONE

Journal Requirements:

Reviewers' comments:

Reviewer's Responses to Questions

**Comments to the Author**

1. Does the manuscript provide a valid rationale for the proposed study, with clearly identified and justified research questions?

Reviewer #2: Yes

2. Is the protocol technically sound and planned in a manner that will lead to a meaningful outcome and allow testing the stated hypotheses?

Reviewer #2: Yes

3. Is the methodology feasible and described in sufficient detail to allow the work to be replicable?

Reviewer #2: Yes

4. Have the authors described where all data underlying the findings will be made available when the study is complete?

Reviewer #2: Yes

5. Is the manuscript presented in an intelligible fashion and written in standard English?

Reviewer #2: No

6. Review Comments to the Author

You may also provide optional suggestions and comments to authors that they might find helpful in planning their study.

Reviewer #2: The aim of this study is to understand how a previous SARS-CoV-2 infection influences the occurrence of perioperative complications. The paper is interesting in all parts but I suggest to include the following paper: 10.2217/bmm-2020-0201

Finally, I suggest to revise the language including avoiding use of elderly and using older adults.

7. PLOS authors have the option to publish the peer review history of their article (what does this mean? ). If published, this will include your full peer review and any attached files.

**Do you want your identity to be public for this peer review?** For information about this choice, including consent withdrawal, please see our Privacy Policy .

Reviewer #2: No

---

## [Author Response · Author response to Decision Letter 2]

10 Apr 2025

Reviewer #2

Thank you for taking the time to thoroughly review our manuscript and for providing such constructive feedback.

Comment #1.1

The aim of this study is to understand how a previous SARS-CoV-2 infection influences the occurrence of perioperative complications. The paper is interesting in all parts but I suggest to include the following paper: 10.2217/bmm-2020-0201

Finally, I suggest to revise the language including avoiding use of elderly and using older adults.

Thank you very much for this comment. We've addressed your concerns about the language and made the needed adjustments. We have taken note of your valuable reference to the study that identified cardiovascular disease as a biomarker for severe coronavirus 2019 (SARS-CoV-2) infection with related poor prognosis. We have taken note of this document with great interest and will be able to use this paper very well in the analysis and discussion in our following paper.

JOURNAL Requirements

Methods: add reference to the STROBE checklist

We have added a reference to STROBE-Checklist.

Title: add “study protocol for a case-control study”

This study follows a case-control design and is conducted both retrospectively and prospectively as a non-interventional observational study.

Methods: Discuss how COVID infection will be ascertained

We have now added to the methods that our patients are only questioned, and that no laboratory evidence of a past infection is required.

Methods: Discuss whether any propensity score matching or attempts of addressing confounders through adjustment will be conducted.

When analysing the data, we will use statistical methods to take confounders into account. We have added this note under Statistical analyses. The exact methods used will be listed in the corresponding paper in the analysis.

Terminology change

We changed the terminology of “elderly”.

Citations for Numerical Rating Scale, PONV scales and delirium and specify the tool in methods text

NRS for describing pain intensity can be found in the study design section. It also describes which number can be assigned to which degree of pain. We have added an appropriate citation.

The PONV score (Apfel-Score) has been added to the methods to show that it is used for risk classification of a patient preoperatively. The explanation of how the Apfel score is calculated has been expanded. A corresponding citation has been added, just as with the PONV-Intensity Scale.

The methods we use to screen for delirium are also shown in the method text below Table 2. For normal wards we use the 3D-CAM and the 4AT and in the ICU we also use the CAM-ICU. Citations are listed for all 3 instruments.

To remain in the correct order, the ranking of the citations has been adjusted.

---

## [Editor Report · Decision Letter 2]

11 Apr 2025

Impact of prior SARS-CoV-2 infection on perioperative cardiac, pulmonary and neurocognitive complications in older patients: study protocol for an observative case control study

PONE-D-24-39467R2

Dear Dr. Saller,

We’re pleased to inform you that your manuscript has been judged scientifically suitable for publication and will be formally accepted for publication once it meets all outstanding technical requirements.

Kind regards,

Kiyan Heybati

Academic Editor

PLOS ONE
---

## [Editor Report · Acceptance letter]

PONE-D-24-39467R2

PLOS ONE

Dear Dr. Saller,

I'm pleased to inform you that your manuscript has been deemed suitable for publication in PLOS ONE. Congratulations! Your manuscript is now being handed over to our production team.

Kind regards,

on behalf of

Dr. Kiyan Heybati

Academic Editor

PLOS ONE